# Identifying the Spatial Architecture That Restricts the Proximity of CD8^+^ T Cells to Tumor Cells in Pancreatic Ductal Adenocarcinoma

**DOI:** 10.3390/cancers16071434

**Published:** 2024-04-07

**Authors:** Yihan Xia, Junrui Ma, Xiaobao Yang, Danping Liu, Yujie Zhu, Yanan Zhao, Xuefeng Fei, Dakang Xu, Jing Dai

**Affiliations:** 1Department of Laboratory Medicine, Ruijin Hospital, Shanghai Jiao Tong University School of Medicine, Shanghai 200025, China; xiayihan@sjtu.edu.cn (Y.X.); 53586mjr@sjtu.edu.cn (J.M.); yangxiaobao@sjtu.edu.cn (X.Y.); liudanping0626@sjtu.edu.cn (D.L.); zhuyujie711@sjtu.edu.cn (Y.Z.); ynzhao@shsmu.edu.cn (Y.Z.); feixuefengyx@163.com (X.F.); 2College of Health Sciences and Technology, Shanghai Jiao Tong University School of Medicine, Shanghai 200025, China

**Keywords:** CODEX, cellular neighborhoods, spatial architecture, proximity, PDAC, anti-tumor immunity

## Abstract

**Simple Summary:**

The proximity to tumor cells is pivotal to the anti-tumor functions of CD8^+^ T cells, but the mechanism underlying the regulation of CD8^+^ T cell spatial distribution remains elusive. Here, we utilize the cellular neighborhood algorithm to identify the spatial architectures that regulate the localization and inter-cellular communication of CD8^+^ T cells in human pancreatic ductal adenocarcinoma. The presence of CD8^+^ T cells, CD4^+^ T cells, and other lymphocytes in the same cellular neighborhoods were identified as one type of spatial architecture that restricted the proximity of CD8^+^ T cells to tumor cells and heralded a poor prognosis. In such architecture, CD8^+^ T cells tended to aggregate around themselves and CD4^+^ T cells instead of approaching tumor cells. In this study, we identified a spatial architecture for the regulation of CD8^+^ T cells and deciphered a novel immune evasion mechanism of pancreatic ductal adenocarcinoma in a topologically regulated manner.

**Abstract:**

The anti-tumor function of CD8^+^ T cells is dependent on their proximity to tumor cells. Current studies have focused on the infiltration level of CD8^+^ T cells in the tumor microenvironment, while further spatial information, such as spatial localization and inter-cellular communication, have not been defined. In this study, co-detection by indexing (CODEX) was designed to characterize PDAC tissue regions with seven protein markers in order to identify the spatial architecture that regulates CD8^+^ T cells in human pancreatic ductal adenocarcinoma (PDAC). The cellular neighborhood algorithm was used to identify a total of six conserved and distinct cellular neighborhoods. Among these, one unique spatial architecture of CD8^+^ T and CD4^+^ T cell-enriched neighborhoods enriched the majority of CD8^+^ T cells, but heralded a poor prognosis. The proximity analysis revealed that the CD8^+^ T cells in this spatial architecture were significantly closer to themselves and the CD4^+^ T cells than to the tumor cells. Collectively, we identified a unique spatial architecture that restricted the proximity of CD8^+^ T cells to tumor cells in the tumor microenvironment, indicating a novel immune evasion mechanism of pancreatic ductal adenocarcinoma in a topologically regulated manner and providing new insights into the biology of PDAC.

## 1. Introduction

CD8^+^ T cells are crucial contributors to anti-tumor immunity. The anti-tumor function of CD8^+^ T cells requires proximity to tumor cells for the recognition of tumor-associated antigens and cytotoxic killing through direct contact [1,2,3,4,5]. It has been reported that the presence of CD8^+^ T cells in the juxtatumoral region of pancreatic ductal adenocarcinoma (PDAC) is associated with a favorable prognosis [5]. However, CD8^+^ T cells rarely infiltrate into PDAC, and most of the intratumor CD8^+^ T cells are excluded from the juxtatumoral region [6]. The tumor microenvironment (TME) of PDAC excludes CD8^+^ T cell infiltration and suppresses the eradication driven by CD8^+^ T cells, promoting tumor growth and metastasis [6,7]. Current studies have focused on the infiltration level of CD8^+^ T cells in the TME, while further spatial information, such as spatial localization and inter-cellular communication, is yet to be defined.

With the development of techniques for multiplex imaging and spatial transcriptomics analysis, there are opportunities to decipher cell–cell interactions in situ and identify critical tissue architectures in tumors, such as the tertiary lymphoid structure which regulates CD8^+^ T cells [8,9,10,11,12,13]. Aside from those previously reported immune-suppressive factors, such as hypoxia and acidification, the function of CD8^+^ T cells has been found to be under the regulation of spatial determinants [6,11,14,15,16,17,18]. The desmoplastic stroma in the TME of PDAC was recently reported to be a key player in the physical restriction of the proximity of CD8^+^ T cells to tumor cells [17,18,19]. After ablation of FAP^+^ cancer-associated fibroblasts, the stroma of PDAC was reshaped to be permissive to T cell infiltration and extravasation, contributing to the efficacy of CAR-T therapy [18]. C. M. Schurch et.al. reported a suppressive program involving Treg cells and CD8^+^ T cells and decreased the enrichment of proliferating Ki-67^+^CD8^+^ T cells in T cell-enriched cellular neighborhoods, indicating that CD8^+^ T cells interact with other cells in specific spatial architectures and their functions were subsequently regulated [9]. Such a phenomenon suggested that spatial localization and inter-cellular communication are key determinants of the function of CD8^+^ T cells. However, the mechanism underlying the regulation of CD8^+^ T cell spatial distribution remains elusive.

In recent decades, the development of analyzing techniques at single cell resolution, such as single-cell RNA sequencing (scRNA-seq) and mass cytometry (CyTOF), has allowed for the unprecedented in-depth understanding of cell types from transcription to immunophenotype in the TME [11,20]. However, these techniques are incapable of identifying and visualizing the diversity of cell types of in situ spatial architectures in the TME. Advances in multiplexed imaging technologies have portrayed both the phenotypes and spatial localization of single cells in the TME [9,10,11,21]. Co-detection by indexing (CODEX) is a recently developed highly multiplexed tissue imaging technique that can spontaneously detect up to 60 cell marker expressions at a single-cell resolution [8,12]. By coupling cell marker expression and tissue localization, the dynamics of inter-cell communication and formation of spatial architectures that are informative for clinical diagnosis within the TME have been identified using spatial analyzing algorithms [9,22,23,24]. The cellular neighborhood algorithm coordinated cell types and tissue regions to which they belong, subsequently dissecting tissues into multiple types of cellular neighborhoods (CNs) [9]. Each identified CN represents one unique type of spatial architecture that plays an integral role in the TME. With the combination of multiplex imaging and computational methods, tertiary lymphoid structures (TLS) were identified as an example of spatial architecture that modulates the function of CD8^+^ T cells [9,11].

By integrating the spatial information of cell types in the TME, new spatial architectures can be found to modulate the distribution of CD8^+^ T cells. In this study, we designed multiplex immunofluorescence with CODEX panels to simultaneously identify PDAC epithelium, T cells, B cells, macrophages and endothelia cells in the PDAC TMA. Coupling digital pathology with spatial analyses, we quantified immune cells density, distribution into tumoral and stromal regions, spatial proximity to other cells, and spatial architectures to which these cells belonged. We defined a total of six conserved and distinct “cellular neighborhoods (CN)” and one unique spatial architecture with CD8^+^ T and CD4^+^ T cell-enriched neighborhoods leads to less proximity of CD8^+^ T cells to the tumor cells associated with shorter overall survival (OS). Our data indicated that neighborhood patterns of immune cells differentiated patient outcomes, and could facilitate future precise medicine in oncology.

## 2. Materials and Methods

### 2.1. Patient Samples Collection and Tissue Microarray Preparation

We collected the PDAC tissues of patients from Ruijin Hospital, of the Shanghai Jiao Tong University School of Medicine (Shanghai, China). We obtained informed consent from all of the participants who were involved (ID: 2021–207). The human ethics committees of Ruijin Hospital affiliated to the Shanghai Jiao Tong University School of Medicine approved this study. The tumor samples were preserved in a formalin-fixed and paraffin-embedded (FFPE) manner. Before the construction of tissue microarray (TMA), the hematoxylin and eosin (H&E) staining results of resected tumor samples were evaluated by a pathologist. Subsequently, FFPE tissue samples from 64 patients were selected and arranged in TMA blocks with diameters of 1.5 mm. The clinical information of each patient can be assessed in Appendix A, and the patients’ basic characteristics can be assessed in Appendix A.

### 2.2. Multiplex CODEX Staining and Image Acquisition

The protocol for CODEX multiplexed imaging used in this study was established previously. [8]. In brief, the deparaffinization of the sections was performed with three washes of xylene (5 min each) after being heated in a 60 °C oven for 1 h and the following rehydration was conducted using a gradient of ethanol solutions (100%, 95%, 85%, 50%). Heat-induced epitope retrieval was performed under 95 °C for 10 min using a pH 9 Tris-EDTA buffer. An antibody cocktail of seven antibodies were designed for the staining of sections (Appendix A). Fluorochrome conjugated reported oligonucleotides were hybridized on 7-cycle experiments with the Akoya PhenoCycler (Akoya Biosciences, Marlborough, MA, USA). The design of cycles of CODEX multiplex staining was listed in Appendix A.

### 2.3. Image Analysis

We utilized HALO image analysis software (version 3.6.4134; Indica Labs, Inc., Albuquerque, NM, USA) to segment tissue and define the phenotypes of cells. DAPI was used for nuclear staining and cytoplasmic markers CD3, CD4, CD8, CD20, CD31, CD68, and PanCK were introduced to identify different cell types. Positive thresholds for each marker were determined based on the fluorescent intensities and all tissue samples were examined. The cell counts, densities, and X, Y coordinates in the center of a single cell were obtained and the graphic images were generated by HALO image analysis platform. Uniform Manifold Approximation and Projection (UMAP) was conducted to visualize and validate the cell phenotyping [25].

### 2.4. Cellular Neighborhood Analysis

A cellular neighborhood analysis was performed as previously described [8]. Neighbors were computationally defined based on the cell-type composition of the ten nearest neighboring cells. For each cell, a window anchoring the central cell and nine other nearest cells was obtained as measured by the Euclidean distance between the X/Y coordinates. The frequency of each cell type was counted among all of the windows, and the windows were then clustered using MiniBatchKmeans clustering with k = 10. The construction of the windows and window clustering were both conducted using the sklearn module in Python (version 3.11.3). (Appendix A). Each cell was then allocated to 10 dominant CNs. Furthermore, the CNs that shared similar biological identities were merged [26]. To validate the CN assignment, these allocations were aligned to the original tissue fluorescent images. Each identified cellular neighborhood was assigned a label based on the cluster map generated from the scaled enrichment score. As the study design, we conducted a preliminary experiment on a test cohort of 10 patients to optimize the parameters of the cellular neighborhood analysis and detect CNs in PDAC (Appendix A). An additional 64 independent patients were enrolled as a validation cohort to validate the above CNs, explore their associations with patient prognosis and carry out further spatial analysis. All of the analyses were performed using Python (version 3.11.3), and the code of the Cellular Neighborhood Analysis can be assessed from https://github.com/nolanlab/NeighborhoodCoordination, (accessed on 13 August 2023).

### 2.5. Calculation of Spatial Distances and Spatial Score between Cell Types

The spatial score was calculated as established before. In brief, the X/Y coordinates for each cell type were determined during the image analysis. The distance from one central cell to the nearest cell with an identity of interest and the averages of these minimal distances per tissue spot were calculated using the shiny app of the spatial score in R (version 4.2.1). The equation for the spatial score calculation is described below:Spatial Score = (distance from central cell to its nearest cell with identity A)/(distance from central cell to its nearest cell with identity B)

### 2.6. Statistical Analysis

Statistical analyses were performed using GraphPad Prism 9.5. to determine statistically significant differences in unpaired data. The statistically significant differences in unpaired data were determined by two-tailed Student’s t-tests and paired *t*-tests were used in paired data. For the multigroup analysis, a one-way analysis of variance (ANOVA) with a Bonferroni multiple comparison test was used. A Pearson correlation was used to determine correlations in the unpaired group. The Kaplan–Meier method was used for the estimation of overall survival outcomes among the subgroups, and log-rank tests were used to determine the statical significance. *p* values or adjusted *p* values < 0.05 were considered to be significant (* *p* < 0.05; ** *p* < 0.01; *** *p* < 0.001).

## 3. Results

### 3.1. CODEX Multiplex Imaging Characterized the Landscape of the PDAC Microenvironment

The TME of PDAC comprised various cell types with different functions. A CODEX panel with seven protein markers was designed to identify these cell types. We performed a seven-marker CODEX on one TMA that incorporated random TME regions from the tumor samples for each patient (Figure 1A). After quality control and spectral unmixing, the multiplex imaging data of 64 regions were obtained (Figure 1B).

The scanned original CODEX images were proceeded by imaging analysis to carry the further spatial analysis. The first step is spectral unmixing to compensate for the overspill of the emission spectrum, adjusting the florescence intensity and setting the threshold for the positive signals (Figure 1C step 1). After compensation of florescence intensity, the microenvironment of PDAC was segmented into a tumor region (PanCK^+^ DAPI^+^ region, red region in Figure 1C step 2), a stroma region (PanCK^−^ DAPI^+^ region, green region in Figure 1C step 2), and a blank region (PanCK^−^ DAPI^−^ region, orange region in Figure 1C step 2) using the HALO algorithm (Figure 1C step 2). Upon protein marker and 4′,6-diamidino-2-phenylindole (DAPI) expression, the HALO algorithm was used to identify each single cell and determine the coordinates of each cell (Figure 1C step 3).

To further annotate the single cells identified in the TME, cell phenotyping was conducted using the HALO algorithm (Figure 1C step 4). A total of eight cell types were defined based on the marker expression and DAPI for a counterstain to identify the cell nucleus and localize the cell. Generally, tumor cells, immune cells, and stromal cells were defined. PanCK^+^ cells were identified as tumor cells. CD31^+^ cells were identified as endothelia cells. CD3 was indicated by the lineage of T cells (CD8^+^ T cells, CD4^+^ T cells, and other T cells whose phenotype is CD3^+^CD8^−^CD4^−^). CD68 indicated the lineage of macrophages. CD20 indicated the lineage of B cells. Those cells with positive DAPI staining but no other positive markers were considered to be stroma components. We additionally conducted UMAP projection to visualize the phenotyping of cell types, apart from tumor cells and stroma cells detected in TME, which showed a result consistent with the cell phenotyping by the HALO algorithm, suggesting a high specificity of markers (Appendix A).

By combining the cellular phenotypes and spatial coordinates, the cellular neighborhood algorithm was used to decipher the inter-cellular communication and spatial co-localization of cell types. For each cell, a window contained the central cell and nine other nearest cells was identified as a local cellular pattern (Figure 1C step 5). A cellular neighborhood algorithm analysis was performed on the multiplex imaging data, and we summarized the local cellular pattern around each individual cell into distinct and conserved cellular neighborhoods (CNs) as an identity (Figure 1C step 6). Finally, the Voronoi diagram of CNs was aligned with the fluorescent CODEX images to validate and interpret the biological function of these CNs (Figure 1C step 7).

### 3.2. The Cellular Neighborhood Algorithm Revealed Distinct Functional Spatial Architectures

As the results of cellular neighborhood analysis returned, we initially identified 10 CNs and found that some of them shared a similar property (Appendix A). Those similar CNs were merged as described in the related Methods section, and a total of six CNs that were biologically interpretable were defined (Figure 2A). The neighborhood enrichment analysis identified the differential enrichment of components within specific cellular neighborhoods. The score revealed the respective frequencies of each cell type within each CN and was subsequently utilized to define the components within specific CNs according to a previous study [9].

Each type of CN represents one specific pattern of local cellular communication and has a stable identity along with the variation of individual cells within it. At a tissue wide level, the TME was dissected into distinct functional spatial architectures formatted by CNs. To further interpret the function of these CNs, we aligned the Voronoi diagram of CNs with the fluorescent CODEX images to reveal the association between the CNs and spatial architectures in the TME (Figure 2B). The tumor region was represented by CN2 that primarily comprised tumor cells and was defined as the ‘bulk tumor’. In the stroma region, immune components and non-immune components were represented by CNs that contained various cell types. For the immune components, CN3 represented the enrichment of lymphocytes, and CN5 was a macrophage niche that was predominantly composed of macrophages and some other cell types. Regarding the non-immune stromal components, three CNs were discriminated: (1) CN4, a perivascular region with enrichment of endothelia cells; (2) CN1 was considered as a pure stroma barrier for the predominant enrichment of stroma cells and the exclusion of other cell types; and (3) CN6 was a reactive stroma barrier that was a mix of stroma cells with immune cells.

### 3.3. Lymphocyte Enrichment and the Reactive Stroma Barrier Heralded a Poor Prognosis

After defining the function of the CNs, we examined the association between the frequency of CNs and the outcome of patients (Figure 3A,B). Those patients with high frequencies of CN3 or CN6 showed significantly shortened overall survival times, while the frequency of other CNs showed no impact on the prognosis. Furthermore, by comparing the frequency of the CNs between patients with short-term survival (<1 year) and long-term survival (>2 years) in our cohort, patients with short-term survival showed significantly higher frequencies of CN3 and CN6 and comparable frequencies of other CNs (Figure 3C).

Consistent with previous studies, CN6 represented the reactive desmoplastic stroma which suppresses the anti-tumor function of CD8^+^ T cells by restricting the proximity of CD8^+^ T cells to tumor cells, thus leading to a worse prognosis [17,18,19]. Despite it being well-known that high level of infiltrated CD8^+^ T cells would lead to a favorable prognosis, it was unexpected that lymphocyte enrichment would lead to a poor prognosis, and this result led us to further decipher the regulatory role of CN3 on anti-tumor immunity.

### 3.4. Both CD4^+^ T Cells and CD8^+^ T Cells in the Same Spatial Architectures was Associated with a Poor Prognosis

The association between the frequency of CNs in the TME and the outcome of patients suggested the regulation of the local cellular pattern on immune cells. Hence, we further explored which cell types were under regulation in CN3 and CN6. The results showed that CD4^+^ T cells, stroma, and CD8^+^ T cells were the top three components of CN3 (Figure 4A left). In comparison with CN3, CN6 was predominantly occupied with stroma (Figure 4A right). To further examine the regulation of these two CNs on these three cell types, we counted the number of these three cell types that belonged to CN3 or CN6 in each patient (Figure 4B).

The subsequent survival analysis revealed that a higher count of CD8^+^ T cells and CD4^+^ T cells located in CN3 was associated with an unfavorable prognosis for patients, while counts of the other cell types located in CN3 showed no association with the prognosis of PDAC (Figure 4C–E, Appendix A). Within CN6, only a higher count of CD8^+^ T cells showed to be associated with a worse prognosis (Figure 4F–H, Appendix A). Notably, despite there being no statistical significance, the high count of stroma located in CN6 showed a tendency to cause a poor prognosis.

Such results suggested that the regulation of CN6 on CD8^+^ T cells may occur in a stroma dependent manner, and this might act as a physical barrier to suppress CD8^+^ T cells. The impact of CN3 on CD8^+^ T cells required further investigation.

### 3.5. CD8^+^ T Cells Were Restricted in the Lymphocyte Enrichment Region

After identifying the potential regulatory role of CN3 on CD8^+^ T cells, we explored how CN3 impacted the spatial distribution of CD8^+^ T cells. We first examined the location of CD8^+^ T cells in the TME. CD8^+^ T cells were found to be predominantly located in CN3, suggesting that the majority of CD8^+^ T cells were under the regulation of CN3 (Figure 5A). Given that the proximity of CD8^+^ T cells to tumor cells is essential for proper anti-tumor function, we further examined the association between the CN3 frequency and the average distance from CD8^+^ T cells to their nearest tumor cells in each patient (Figure 5B). The correlation analysis showed that a higher frequency of CN3 was associated with a farther distance between CD8^+^ T cells and tumor cells (Figure 5C). CD8^+^ T cells within CN3, compared with CD8^+^ T cells within other CNs, showed a significantly closer distance to their nearest CD8^+^ T cells but a farther distance to their nearest tumor cells (Figure 5D,E).

These data suggested that CD8^+^ T cells were more likely to gather around themselves and were restricted within CN3 instead of approaching tumor cells.

### 3.6. CD4^+^ T Cells Restricted the Proximity of CD8^+^ T Cells to the Tumor in the Lymphocyte Enrichment Region

The previous data suggested that CD8^+^ T cells located in lymphocyte enrichment region were distant from tumor cells. Both the CD4^+^ T cells and CD8^+^ T cells were found to be simultaneously enriched in the lymphocyte enrichment region. We wondered whether CD4^+^ T cells could regulate the proximity of CD8^+^ T cells. Thus, we aimed to explore the spatial relationship between CD8^+^ T cells, CD4^+^ T cells, and tumor cells. A strong negative association between the CN3 frequency and the average distance from CD8^+^ T cells to their nearest CD4^+^ T cells in each patient was found, indicating that CN3 promoted the proximity of CD8^+^ T cells to CD4^+^ T cells (Figure 6A). Similarly, CD8^+^ T cells within CN3 were also found to be closer to CD4^+^ T cells in comparison to CD8^+^ T cells within other CNs (Figure 6B). To further examine the spatial relationship between CD8^+^ T cells, CD4^+^ T cells, and tumor cells, we introduced the spatial score to describe which cell type CD8^+^ T cells were more likely to interact. As illustrated in Figure 6C, we established CD8^+^ T cells as the center cells and calculated the distance to the nearest tumor cells (left) and CD4^+^ T cells (right). The spatial score is the ratio of left to right. A high spatial score indicated the proximity of CD8^+^ T cells to CD4^+^ T cells and vice versa (Figure 6C). Patients with a high frequency of CN3 showed significantly higher spatial scores than those with a low frequency of CN3, suggesting that CD4^+^ T cells suppressed the proximity of CD8^+^ T cells to tumor cells in the lymphocyte enrichment region (Figure 6D).

Regarding CN6, a higher frequency of CN6 was associated with a farther distance between CD8^+^ T cells and tumor cells and a closer distance between CD8^+^ T cells and CD4^+^ T cells, and patients with a high frequency of CN6 showed significantly higher spatial scores than those with a low frequency of CN6 (Appendix A). Despite showing a similar trend to previous results of CN3, in Figure 4C,D,F,G, both of CD8^+^ T cells and CD4^+^ T cells located in CN3 contributed an unfavorable prognosis for patients, while only CD8^+^ T cells not CD4^+^ T cells within CN6 were associated with a poor prognosis for patients, indicating a coordination between CD8^+^ T cells and CD4^+^ T cells in CN3.

## 4. Discussion

Defining tumor immunophenotypes based on the individual cell level does not discriminate clinical outcomes, for the level of density of individual cell types in tissue fails to reflect the biological complexity of tumors. Therefore, the identification of cellular neighborhoods, defined as a collection of stereotyped features of distinct tumor regions, represents an alternative approach to tissue analysis [9]. In this study, we used CODEX to study the immunologic features of patients with PDAC using a TMA, quantifying cells known to be associated with outcomes in patients with PDAC: tumor cells, T cells, B cells, macrophages, endothelia cells and testing their infiltration and localization for associations with clinical characteristics. CODEX multiplex imaging was coupled with a cellular neighborhood analysis to reveal the spatial regulation pattern. After combing digital pathology with spatial analysis, one unique spatial architecture with CD8^+^ T and CD4^+^ T cells enriched neighborhoods is identified and leads to less proximity of CD8^+^ T cells to the tumor cells, heralding a shorter OS. Our data demonstrate that patterns of immune neighborhoods can differentiate patient outcomes and could inform future precision medicine in oncology.

The CODEX imaging is capable of simultaneously visualizing multiple protein markers in situ and enabling a deep view into the single-cell spatial relationships in tissues, which showed strength in quantitative systemic characterization of tissue architecture, and thus, the CODEX multiplex imaging was introduced in this study to identify the spatial architectures in human PDAC [8,9,10]. In comparison with CODEX and other imaging-based methods, the spatial transcriptomic analysis offers high gene expression quantification and insights into molecular regulatory pattern but at a resolution of clusters of ~10–100 cells instead of subcellular resolution, which is not suitable for analyzing small immune populations or clusters and revealing the tissue architectures [8,9,10,11,12,13]. As for the traditional single-cell technologies, such as scRNA-seq and CyTOF, they have allowed for the investigation of cell types from transcription to immunophenotype in the TME while they are not capable of providing spatial information of cell types [9,11].

Details of the CODEX workflow are shown in Figure 1, including image preprocessing, tissue and cell segmentation, cell type identification, spatial analysis, and cell neighborhood analysis. Cells in the tumor microenvironment, although sometimes seemingly random, are also expected to follow a certain “order”. The cellular neighborhood algorithm identifies local cellular communication patterns around each individual cell and summarizes these patterns into CNs. Each identified CN represents one unique type of spatial architecture that plays an integral role in the TME.

We utilized the cellular neighborhood algorithm to dissect the TME of PDAC in order to identify the spatial architectures in the TME of PDAC. We found that the TME of PDAC was composed of a bulk tumor region, a lymphocyte enrichment region, a macrophage niche, a perivascular region, and two types of stroma barriers (a pure stroma barrier and a reactive stroma barrier). Consistent with previous studies, the reactive stroma barrier (CN6) was found to be associated with a poor prognosis. The desmoplastic stroma in the TME of PDAC was recently considered to be a key player to physically restrict the proximity of CD8^+^ T cells to tumor cells [17,18,19]. Unexpectedly, the lymphocyte enrichment region (CN3) was also found to be associated with a poor prognosis in patients with PDAC. Furthermore, a high count of CD8^+^ T cells and CD4^+^ T cells located in CN3 heralded a poor prognosis, suggesting that, despite the high infiltration of lymphocytes, the location of these cells may alter their function.

The TME is a complex of different cell types and the organization of these components will affect the anti-tumor function and spatial distribution of CD8^+^ T cells [6,9,14]. Previous studies have revealed that spatial architectures, such as tertiary lymphoid structure and T cell enriched cellular neighborhoods As for the lymphocyte enrichment region we identified in this study, our data suggested that the CD8^+^ T cells were predominantly restricted in the lymphocyte enrichment region, showing an increased proximity to CD4^+^ T cells and suppressed proximity to tumor cells, thus leading a poor prognosis. Regarding the potential mechanism of the formation of this spatial architecture, it has been previously reported that CD4^+^ T cells could secrete CXCL12, CXCL13, CCL19, CCL21 and LTβ to recruit CD8^+^ T cells, contributing to the aggregation of CD8^+^ T cells and CD4^+^ T cells and maintaining ectopic lymphoid-like structures [27,28,29,30,31].

Intriguingly, the CD8^+^ T cells located within the lymphocyte enrichment region showed an altered spatial behavior. The spatial location and presence of CD8^+^ T cells is pivotal to the anti-tumor immunity response to PDAC [5,32,33]. Some components, such as myeloid cells and desmoplastic components in the TME, have been reported to impact the distribution of CD8^+^ T cells, while the role of local communication patterns around CD8^+^ T cells has yet to be defined [17,18,19,22,34].The proximity of CD8^+^ T cells to tumor cells has been well known to facilitate the recognition of tumor-associated antigens and subsequent tumor eradication [1,2,3,4,5].To assess the proximity of CD8^+^ T cells to tumor cells, we examined the distance from CD8^+^ T cells to their nearest tumor cells. The results showed that those CD8^+^ T cells within CN3 showed a farther distance compared to CD8^+^ T cells within other CNs. Instead of approaching tumor cells, CD8^+^ T cells within CN3 showed a significantly closer distance between the nearest CD8^+^ T cells and CD4^+^ T cells than the other CD8^+^ T cells. It was found that CD8^+^ T cells were restricted in the lymphocyte enrichment region rather than infiltrated into the juxtatumoral region. The current data were not sufficient for us to discover the reason why CD8^+^ T cells tended to gather around themselves or CD4^+^ T cells. Here, our study showed the results as a proof of concept to reveal cellular neighborhoods in the PDAC tumor tissues and their associations with patient prognosis. If a simplified workflow to assist researchers in the field or clinical diagnosis could be established in future study, it would be a valuable resource for the scientific community.

## 5. Conclusions

In this study, we utilized a cellular neighborhood analysis to decipher the TME of PDAC based on CODEX multiplex imaging. We identified one unique spatial architecture with the presence of CD8^+^ T cells, CD4^+^ T cells, and other lymphocytes in the same region that restricted the proximity of CD8^+^ T cells to tumor cells, eventually leading to a worse prognosis. The identified spatial architecture featured lymphocyte enrichment, and this may be detected by a clinical pathologist using other approaches, serving as a predictive biomarker for precise diagnosis (Figure 7).

## Figures and Tables

**Figure 1 cancers-16-01434-f001:**
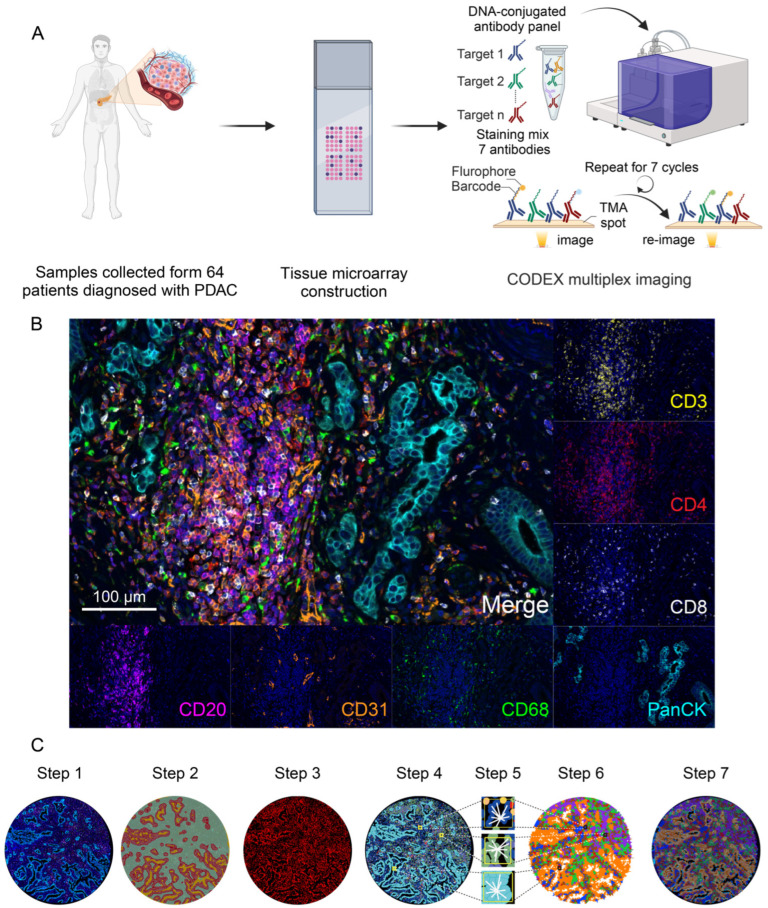
Workflow of CODEX Multiplex Imaging on PDAC Tissue and Imaging Analysis. (**A**) Scheme plots of the tissue microarray construction and CODEX staining. Created with BioRender.com. The publication license is available in Appendix A. (**B**) Representative images of the CODEX multiplex staining with seven protein markers. (**C**) Scheme plots for the imaging analysis and spatial analysis procedure. The scanned original CODEX multiplex imaging was proceeded by spectrum unmixing (step 1); tissue region segmentation (step 2); cellular segmentation; (step 3); cellular phenotyping; (step 4); local cellular pattern identification (step 5); cellular neighborhood clustering (step 6); and tissue alignment (step 7).

**Figure 2 cancers-16-01434-f002:**
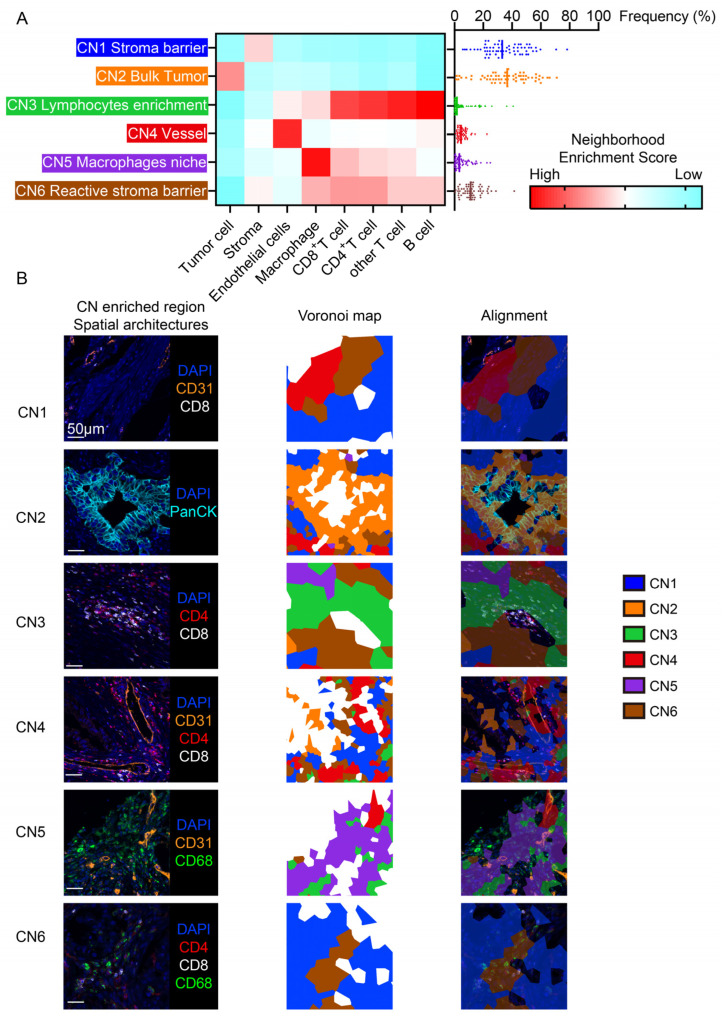
Characterization of the cellular neighborhoods and their alignment to spatial architectures. (**A**) Identification of six conserved CNs based on eight distinct cell types in the 64 PDAC samples and their respective proportion of all CNs per sample. (**B**) Representative multiplex imaging of spatial architectures and alignment of the CN Voronoi diagram.

**Figure 3 cancers-16-01434-f003:**
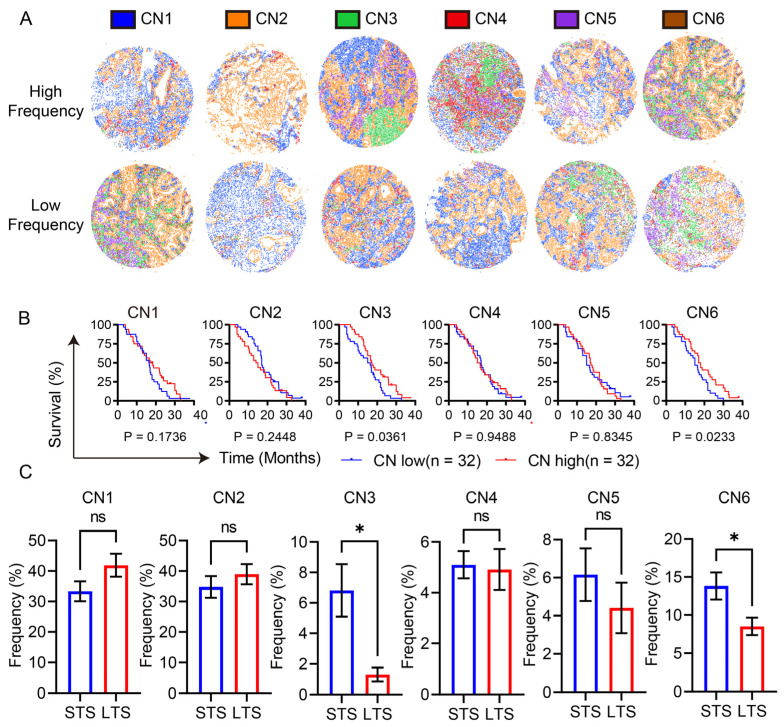
Identification of two unique spatial architectures that heralded a poor prognosis. (**A**) Representative Voronoi diagram illustration of patients with high or low frequencies of CNs. Each color in the Voronoi diagram represents one type of CN. (**B**) Survival plots of the CN frequencies. The blue lines represent the survival curve of the low CN frequency group, and the red lines represent the survival curve of the high CN frequency group. (**C**) Comparison of the CN frequencies between patients with short-term survival (*n* = 19) and long-term survival (*n* = 12). Student’s *t* test was conducted. * *p* < 0.05 STS stands for short-term survival and LTS stands for long-term survival.

**Figure 4 cancers-16-01434-f004:**
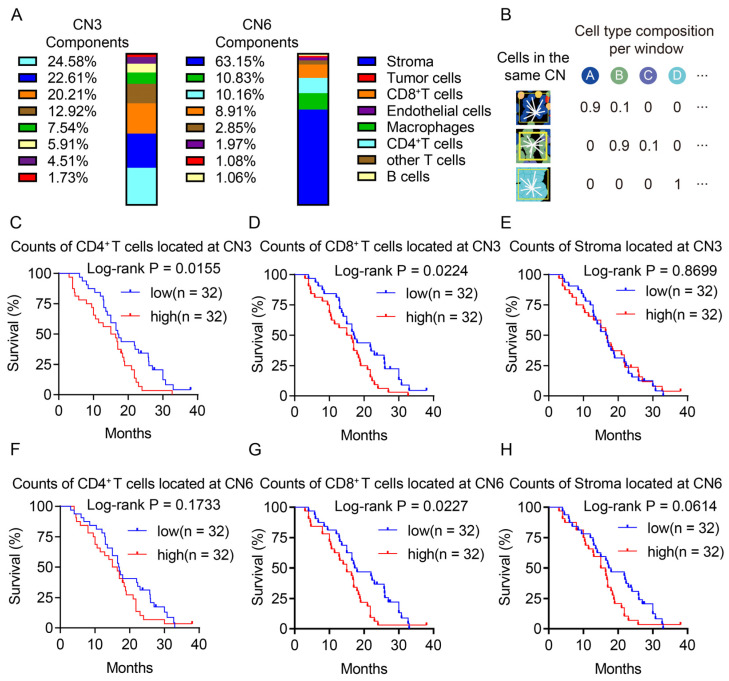
High count of CD4^+^ T cells and CD8^+^ T cells located in the same spatial architectures heralded a poor prognosis. (**A**) Bar charts of the CN3 and CN6 components of all of the patients. Each color in the bar chart represents one cell type. (**B**) Illustration of the cell type localization in the CNs. (**C**–**H**) Survival plots of the counts of CD8^+^ T cells, CD4^+^ T cells, and stroma belonging to CN3 or CN6. The blue lines represent the survival curve of the low-count group, and the red lines represent the survival curve of the high-count group.

**Figure 5 cancers-16-01434-f005:**
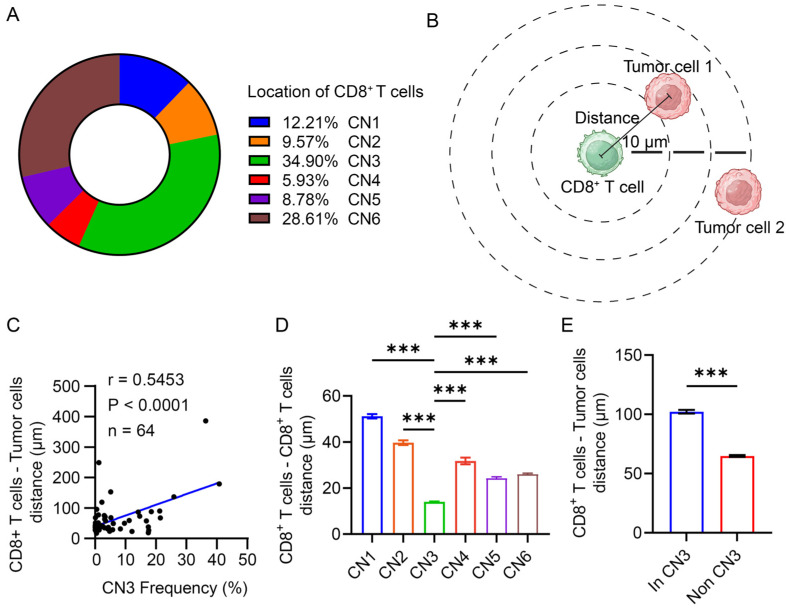
CD8^+^ T cells were restricted in the lymphocyte enrichment region. (**A**) Pie chart of the CD8^+^ T cell locations of all of the patients. Each color in the pie chart represents one type of CN. (**B**) Graphic illustration of the measurement of the distance between CD8^+^ T cells and tumor cells. The black line indicates the distance from each CD8^+^ T cell to the nearest tumor cell. Created with BioRender.com. The publication license is available in Appendix A. (**C**) Correlation scatter plot of the CN3 frequency and the average distance from each CD8^+^ T cell to the nearest tumor cell in each patient. (**D**) Comparison of the distance from each CD8^+^ T cell to the nearest tumor cell within each CN. The different colors represent CD8^+^ T cells from the different CNs. One-way ANOVA test was conducted. *** *p* < 0.001. (**E**) Comparison of distance from each CD8^+^ T cell to the nearest tumor cell within CN3 or non-CN3. Student’s *t* test was conducted. *** *p* < 0.001.

**Figure 6 cancers-16-01434-f006:**
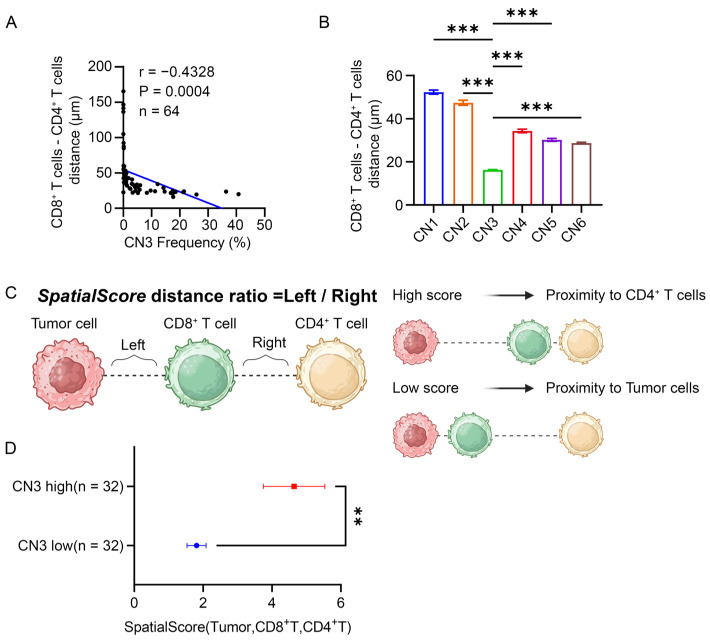
CD8^+^ T cells were closer to CD4^+^ T cells than to tumor cells. (**A**) Correlation scatter plot of the CN3 frequencies and average distance from each CD8^+^ T cell to the nearest CD4^+^ T cell in each patient. (**B**) Comparison of the distance from each CD8^+^ T cell to the nearest CD4^+^ T cell within each CN. The different colors represent the CD8^+^ T cells from different CNs. One-way ANOVA test was conducted. *** *p* < 0.001. (**C**) Graphic illustration of the calculation of the CD8^+^ T cells centered spatial score between CD4^+^ T cells and tumor cells. A high spatial score indicated the proximity of CD8^+^ T cells to CD4^+^ T cells, and a low spatial score indicated the proximity of CD8^+^ T cells to tumor cells. Created with BioRender.com. The publication license is available in Appendix A. (**D**) Comparison of the spatial scores between patients with high CN3 frequencies or low CN3 frequencies. Student’s *t* test was conducted. ** *p* < 0.01.

**Figure 7 cancers-16-01434-f007:**
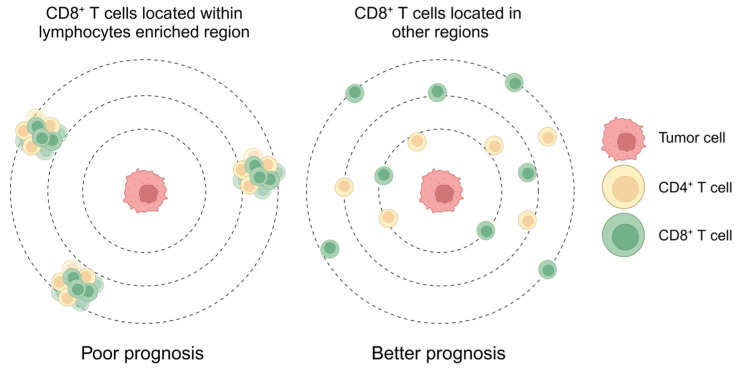
Spatial architecture that enriched with CD8^+^ T cells and CD4^+^ T cells restricted the proximity of CD8^+^ T cells to tumor cells and heralded a poor prognosis. Created with BioRender.com. The publication license is available in Appendix A.

## Data Availability

The authors declare that the main data supporting the findings of this study are available within the article and Appendix A. The data presented in this study are available upon request from the corresponding author. The code of the Cellular Neighborhood Analysis can be assessed from https://github.com/nolanlab/NeighborhoodCoordination, (accessed on 13 August 2023).

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
