# Peer review of "Identifying the Spatial Architecture That Restricts the Proximity of CD8+ T Cells to Tumor Cells in Pancreatic Ductal Adenocarcinoma"

_cancers, 2024, doi:10.3390/cancers16071434_

Round 1

Reviewer 1 Report

Comments and Suggestions for Authors

The manuscript investigates the tumor microenvironment (TME) of pancreatic ductal adenocarcinoma (PDAC) by combining CODEX multiplex imaging and cellular neighborhood analysis. The study reveals distinct spatial architectures within the TME characterized by different cell types and their interactions. Six cellular neighborhoods (CNs) are identified, including a bulk tumor region, lymphocyte enrichment region, macrophage niche, perivascular region, and two types of stroma barriers. Surprisingly, the lymphocyte enrichment region (CN3) is associated with a poor prognosis in PDAC patients, indicating altered immune cell function within this spatial architecture. Further investigation reveals high counts of CD8+ T cells and CD4+ T cells within the lymphocyte enrichment region. Specifically, CD8+ T cells exhibit restricted proximity to tumor cells and this restriction is influenced by the presence of CD4+ T cells, suggesting a regulatory role in immune cell localization. Overall, the manuscript underscores the significance of spatial analysis in understanding the TME and its impact on patient outcomes. Additionally, it provides insights into the complex interplay between immune cells and the TME in PDAC.

1. It would strengthen the conclusions by conducting additional validation studies using independent datasets or experimental models to confirm the observed associations between cellular neighborhoods (CNs) and patient prognosis.

2. The manuscript suggested correlations between immune cell localization and patient prognosis. However, it needs a clearer potential explanations to discuss the observed associations for causal relationship.

3. It would be beneficial to perform scRNA-seq and spatial transcriptomics analysis to complement the spatial information provided by CODEX imaging. Additionally, t-SNE and  UMAP could be used to visualize and explore the high-dimensional CODEX imaging data. Differential spatial analysis could be performed to identify components within specific CNs.

Reviewer 2 Report

Comments and Suggestions for Authors

The paper “Identifying the Spatial Architecture that Restricts the Proximity of CD8+ T Cells to Tumor Cells in Pancreatic Ductal Adenocarcinoma” designed co-detection by indexing (CODEX) was to characterize PDAC tissue regions with 7 protein markers in order to identify the spatial architectures that regulate CD8+ T cells in human pancreatic ductal adenocarcinoma (PDAC). Here are still some shortcomings that need to be further improved or explained.

Comments:

Q1. The repetition rate of the paper is too high, which needs to be improved.

Q2. Line 63-65, “Such a phenomenon suggested that spatial localization and inter-cellular communication are key determinants of the function of CD8+ T cells. However, the mechanism underlying the regulation of CD8+ T cell spatial distribution remains elusive.” Did the authors mean that the inter-cellular communication of the function of CD8+ T cells could be elucidated clearly? The manuscript did not make any reference to that.

Q3. The information of fluorescent markers is recommended to be supplemented along with immunofluorescence staining antibodies, as well as the wavelength design for photography.

Q4. The figures showed that CD8+ T cells did not seem to be the most abundant cells in the tumor microenvironment, so why choose this kind of cells as the research object? Whether the authors used flow cytometry to determine the proportions of each immune cell subsets? If possible, please supplement related scatter plots.

Q5. The research in this thesis is interesting, but why choose CD31,DAPI, etc. besides CD4 and CD8?

Q6. The structure of this paper is clear, and the addition of schematic diagrams makes us better understand the conclusion, but the effects of schematic diagrams are difficult to analyze in the actual immunofluorescence photos.

Reviewer 3 Report

Comments and Suggestions for Authors

Comments:

1. Are any markers in CD4 and CD8 T cells involved in forming spatial architecture around tumor cells? Such as L-selectin, CD45, CCR7, Leu-3, etc. 

2. Is the anatomical location of the tumors associated with the T cell-spatial formation? For example, head, body, and tail of the pancreas.

3. In Figure 3C: There is a significant difference between STS and LTC in CN3 and CN6. Authors showed the detailed study in CN3. How about CN6? For example, what would be the results in CN6 in Figure 5C and Figure 6A and 6D?

4. In Figure 7: what would be the mechanism to form aggregates in CD4+ and CD8+ T cells in poor prognosis condition?

5. In supplemental figure S2: Do you have more detailed patients' information? BMI, smoking and alcohol history, location of M1, etc. Why uses age 65 as cut-off? Menopausal status in female younger than age of 65?

6. The sample size is relatively small in this study. Please explain. Also the patients are from certain areas of China?

Round 2

Reviewer 2 Report

Comments and Suggestions for Authors

I have no other questions, while the percent match of 34% was still unacceptable.

Author Response

We appreciated your expertise and acknowledgement of our work. We have done a minor revision to reduce the repetition rate.

Reviewer 3 Report

Comments and Suggestions for Authors

No more comments

Author Response

We appreciated your expertise and acknowledgement of our work. We have done a minor revision.